

# A genome-wide investigation of microsatellite mismatches and the association with body mass among bird species

Haiying Fan and Weibin Guo

Department of Ecology, College of Life Sciences, Wuhan University, Wuhan, China

## ABSTRACT

Mutation rate is usually found to covary with many life history traits of animals such as body mass, which has been readily explained by the higher number of mutation opportunities per unit time. Although the precise reason for the pattern is not yet clear, to determine the universality of this pattern, we tested whether life history traits impact another form of genetic mutation, the motif mismatches in microsatellites. Employing published genome sequences from 65 avian species, we explored the motif mismatches patterns of microsatellites in birds on a genomic level and assessed the relationship between motif mismatches and body mass in a phylogenetic context. We found that small-bodied species have a higher average mismatches and we suggested that higher heterozygosity in imperfect microsatellites lead to the increase of motif mismatches. Our results obtained from this study imply that a negative body mass trend in mutation rate may be a general pattern of avian molecular evolution.

## INTRODUCTION

It has long been recognized that the molecular evolutionary rates always covary with many life history traits of animals. Numerous studies have documented a negative relationship between the rate of molecular evolution and body mass (*Nabholz, Glémin & Galtier, 2008*; *Bromham, 2011*; *Amos & Filipe, 2014*), where genes in small-bodied species are likely to evolve faster than those in large-bodied species. This has been readily explained by higher number of mutation opportunities per unit time (generation length hypothesis, *Li et al., 1996*) or higher mutation probability in a round of DNA replication due to higher metabolic rate (metabolic rate hypothesis, *Mindell et al., 1996*) in small-bodied species. Although the precise reason for the pattern is not clear at present, to determine the universality of this pattern, we need to study additional form of genetic mutation besides mitochondrial DNA or nuclear 'genes' which are most frequently used. The first to consider is the fastest evolving components of the genome such as microsatellites.

Microsatellites, also known as simple sequence repeats (SSRs), are tandem repeats of simple nucleotide motifs, which have wide coverage in eukaryotic and prokaryotic genomes (*Tóth, Gáspári & Jurka, 2000*; *Ellegren, 2004*; *Adams et al., 2016*). One feature of

Corresponding author
Haiying Fan,
fanhaiying1989@126.com

microsatellites is that they have a high mutation rate ($10^{-7}$ to $10^{-3}$ mutations per locus per generation), leading to high heterozygosity and extensive length polymorphisms (*Kruglyak et al., 2000*). It has long been assumed that the major cause of variation of microsatellite repeats is replication slippage (*Kornberg et al., 1964*; *Bhargava & Fuentes, 2010*), which will increase or decrease repeat copy numbers in microsatellites. Specifically, when it creates a loop in one of strands, a slippage error occurs. If the loop is formed in the replicating strand, it will introduce an insertion. If the loop is in the template strand, a deletion will emerge. Several mathematical models of microsatellite evolution have been proposed to represent the mutation processes of microsatellites, such as stepwise mutation model (SMM) of *Ohta & Kimura (1973)*, which suggests that mutation in microsatellite loci occurs by one repeat unit at a time.

Many studies on microsatellites have explored the frequencies, abundance and polymorphism of microsatellites in the genomes (*Wang et al., 2014*; *Qi et al., 2015*; *Adams et al., 2016*). Few, if any, have correlated these microsatellite characters to the life history traits of a species. Specifically, microsatellites are hypothesized to experience a life cycle: start short (birth) and expand predictably due to mutation bias (expansion) until they become unstable and either collapse or degrade through internal point mutations (contraction and death) (*Chambers & MacAvoy, 2000*; *Buschiazzo & Gemmell, 2006*). Life history traits of species are expected to have an influence on the life cycle —'birth', expansion, and 'death'—of microsatellites in the genome (*Amos & Filipe, 2014*). For example, in smaller species, higher mutation rate allows the 'birth' and expansion of microsatellites faster, due to higher mutation rate and slippage rate. Since the death rate is lower than the birth rate, microsatellites tend to accumulate in the genome (*Buschiazzo & Gemmell, 2006*). In that, the smaller species harbour a higher frequency of microsatellites across the genome, which has been proved in mammals (*Amos & Filipe, 2014*).

It is well known that except for repeat copy number variation, a microsatellite (e.g., ATATATATAT) also suffers from nucleotide substitutions and insertion/deletion mutations, hence becoming imperfect (e.g., ATATATCATAT: AT repeat with an insertion of C). Perfect and imperfect microsatellites are thus defined. It has been found that genomes possess a relatively small but significant number of imperfect microsatellites (*Brinkmann et al., 1998*). Mismatch variation of imperfect microsatellites is critical for their maintenance in the genome and imperfect microsatellites are more stable compared to perfect microsatellites since the former is less prone to slippage mutation (*Sturzeneker et al., 1998*). Several previous studies have already revealed the genome-wide motif imperfection pattern among species (e.g., *Behura & Severson, 2015*). Nevertheless, our understanding of motif mismatches in imperfect microsatellites is still very limited and their correlation with life history traits remains to be revealed and explained.

In this study, we used 65 avian genome sequences, employing SciRoKo (*Kofler, Schlötterer & Lelley, 2007*) to search SSRs in the whole genome. We chose avian genomes for this study because microsatellites have been widely used in population genetics of bird species, yet the pattern of microsatellites mismatches in birds is still not well understood, mostly owing to the lack of avian genomic information. With the advance of whole genome sequencing, evolution of microsatellites is attracting attention from researchers. With the

genome-wide microsatellites data in hand, we presented the first detailed comparative study of microsatellites, aiming to reveal the patterns of motif mismatches across different bird species and to help understand the relationship with life history traits.

## MATERIALS & METHODS

### Genome sequences and body mass

We downloaded FASTA files of the 65 avian genomes from NCBI and GigaDB (http://dx.doi.org/10.5524/101000). These avian species represent nearly all of the major clades of living birds. We compiled data from the original and secondary references and the world-wide web about the mean body mass of adult males and females (Table S1). If a mean value was not provided for a species, we took the median of the range. Where separate body masses were given for males and females, the average value of the masses was calculated.

### Identification of microsatellites

We searched microsatellites in each genome sequences using SciRoKo 3.4, a simple sequence repeats (SSRs) identification program (*Kofler, Schlötterer & Lelley, 2007*), with the default parameters (minimum score = 15 and mismatch penalty = 5) in the mismatched modes. In addition, we used different parameters to search SSRs (minimum score = 15 and mismatch penalty = 3, minimum score = 10 and mismatch penalty = 5) considering changing in parameters would affect the results of this study. Specially, the motif mismatches refer to the number of base mismatches of an imperfect microsatellite compared with its idealized perfect counterpart. For example, the string TACTACTAGTACTAC, is a trinucleotide repeat with five repeats and, by comparison with its idealized perfect counterpart (consensus repeat), it has a mismatch of 1. The number of mismatches of each microsatellites as well as their length for each genome was used for different comparative analyses across the species.

### Statistical analyses

In this study, we used phylogenetic generalized least-squares regression (PGLS) (*Freckleton, Harvey & Pagel, 2002*) implemented in the package 'ape' (*Paradis, Claude & Strimmer, 2004*) to control shared ancestry (for the script used, see Fig. S1). We used the evolutionary tree of the 48 bird species estimated by *Jarvis et al. (2014)* as a backbone topology, and used the phylogenetic information provided by *Jetz et al. (2012)* to add the remaining 17 species (for the resulting phylogeny, see Fig. S2). In order to achieve the statistical requirements for linearity and normality, adult average mass were log10-transformed prior to analysis. Average mismatches was reciprocal transformed. GC content was arcsine square root transformed.

Firstly, we computed some basic statistics on characteristics of microsatellite loci in 65 bird genomes (Tables S2–S5). Secondly, to better understand the occurrence of motif mismatches in bird genomes, we determined the frequency of microsatellites of 20 bp that either lacks mismatches or harbours one mismatch for each species (Table S6). 20 bp was used because that the average length of perfect microsatellites of 65 birds is 20 bp. Then we computed the ratio of imperfect (mismatch = 1) repeats frequency to perfect (mismatch

= 0) ones. Then, we employed a PGLS, treating the ratio of imperfect (mismatch = 1) to perfect (mismatch = 0) repeats as a dependent variable and body mass as an explanatory variable. Thirdly, we explored whether or not the extent of motif mismatches is related to genomic abundance of imperfect microsatellites. We first calculated the probability of per-site mismatches (the total number of mismatches divided by total lengths of all loci) in each genomes. Then the expected number of mismatches was determined based on the length and compared with the observed number of mismatches in each imperfect microsatellite (Table S7). The first paired sample $t$-test was conducted between the numbers of microsatellites harbouring more mismatches than expected and that of carrying fewer mismatches than expected. The second paired sample $t$-test was performed between imperfect repeats that have a length of at least 30 bp and have either <3 or ≥3 mismatches (Table S8). Finally, to test whether differences of mismatches in imperfect microsatellites link with body mass, we fitted a PGLS analysis with average mismatches of imperfect SSRs as dependent variable and body mass as a predictor. The average mismatches of imperfect SSRs in individual genomes was estimated as the sum of mismatches divided by the number of imperfect microsatellites (Table S1). Average mismatches was used because it indicates the mismatches in an 'average' imperfect SSR. For controlling the probability that GC content will have a potential influence on microsatellite mismatches, we added it to the models as a predictor variable. Taking di-, tri-, tetra-, penta- and hexamers as the five classes of repeats, we repeated the PGLS analysis in each repeat type. Since the mutations in the mononucleotide repeats tend to cause the emergence of a new motif of other repeat type, we excluded it from our analysis. All statistical analyses were conducted with R 3.1.2 (*R Core Team, 2014*).

## RESULTS

### Characteristic of microsatellite loci in 65 avian species

In total, 11803896 SSR loci with a minimum length of 15 bp were identified from 65 avian genome assemblies, and were classified into mono-, di-, tri-, tetra-, penta- and hexanucleotide SSRs according to the motif length (Table S2). Among these, mononucleotide SSRs are the most abundant (42.3%) type, followed distantly by tetra- (18.8 %) and pentanucleotide SSRs (17.1%) (Table S2; Fig. S3). The SSR abundance composition and SSR density of the birds varies greatly among species, with the maximum value in *Anas platyrhynchos* (416,040 counts; 376.49 counts/Mb) and the minimum value in *Melopsittacus undulatus* (81,643 counts; 73.07 counts/Mb). Additionally, the SSR abundance composition are predicted by genome size ($\beta \pm SE = 1.56 \pm 0.64$, $t = 2.44$, $P = 0.017$, $R^2 = 0.09$).

### Frequency of imperfect microsatellites in bird genomes

The number of imperfect microsatellites varies among the birds species and the imperfect repeats account for 15–27% of all microsatellites searched from the genome assemblies of the 65 bird species as shown in Table S3. The imperfect repeats represented less than 0.2% of the genome sequence in most of these birds except four species (*Anas platyrhynchos, Calypte anna, Columba livia, Picoides pubescens*) (Fig. S4). The data in Table S3 and Fig. S4

shows that the frequency of imperfect microsatellites in bird genomes appears substantial variation among these species. It was observed that *Anas platyrhynchos* has a higher percentage of imperfect microsatellites than other bird species. Moreover, the proportion of imperfect repeats varies differentially among species, to some extent, depending on the motif size of microsatellites. Specifically, the paired sample $t$-test results indicated the di-, tri- and hexanucleotide SSRs have an increased rate of motif mismatches compared with all other types of motif size (Table S4). Furthermore, it seems that this pattern is conserved among different avian species.

### The occurrence of motif mismatches

Imperfect microsatellites are longer than perfect microsatellites in each species (37 vs 20 bp; $t = 33.334$, $df = 64$, $P < 0.001$; Table S5). The PGLS analysis revealed that the small-bodied species has a higher ratio of imperfect (mismatch = 1) to perfect (mismatch = 0) repeats of 20bp than large-bodied species ($\beta \pm SE = -0.006 \pm 0.002$, $t = 2.86$, $P = 0.006$, $R^2 = 0.12$).

### The accumulation of mismatches in imperfect microsatellites and genomic abundance

The paired sample $t$-test revealed that the microsatellites harboring mismatches higher than expected has significantly lower abundance than that carrying mismatches lower than expected (13,381 versus 25,138 counts; $t = 22.651$, $df = 64$, $P < 0.001$; Table S7), implying that the imperfect microsatellites which containing more mismatches have lower abundance in the genome. We also found that loci with three or more number of mismatches are less common than that have less than three mismatches (7,364 vs 18,361 counts; $t = 11.316$, $df = 64$, $P < 0.001$; Table S8).

### Correlation between body mass and average motif mismatches

We found that on a whole genome scale, the average body mass accounts for 28.2% of the variation in average mismatches of imperfect SSRs (Table 1, Fig. 1). Body mass also has a significantly negative correlation with microsatellites mismatches in five motif length classes (Table 1, Fig. 2). This negative correlation remains significant when adding GC content to the regression models. Inclusion of GC content only enhances the model's explanatory power slightly except in tetra- and pentanucleotide SSRs. When we used different parameters including minimum score 15 and mismatch penalty 3 and a minimum score of 10 and mismatch penalty 5 to search microsatellites in the genomes, the results of repeated analyses were highly consistent (Table S9). This confirmed that our observations were not influenced by the search parameters of microsatellites.

## DISCUSSION

In the present study, we did a genome-wide search of microsatellites using SciRoKo with the same parameters to ensure that the program can search all possible microsatellites with the same probability for every genome. Microsatellites search results showed that the frequency of microsatellites varies extensively among species. We have also found a positive relationship between microsatellites abundance and genome size among 65 bird species, which is consistent with earlier studies (e.g., *Hancock, 1996*). After providing a

**Table 1** Result for the relationship between average mismatches and body mass fitted in PGLS analyses.

| | | | Coefficients | | | | | |
| | | | Body mass | | | GC content | | |
| Type | Model | $R^2$ | $\beta \pm SE$ | $t$ | $P$ | $\beta \pm SE$ | $t$ | $P$ |
|---|---|---|---|---|---|---|---|---|
| All | BM | 0.282 | $0.015 \pm 0.003$ | 4.974 | <0.001 | | | |
| | BM + GC | 0.340 | $0.015 \pm 0.003$ | 5.102 | <0.001 | $-2.533 \pm 1.090$ | 2.324 | 0.023 |
| Di | BM | 0.279 | $0.022 \pm 0.005$ | 4.932 | <0.001 | | | |
| | BM + GC | 0.332 | $0.022 \pm 0.004$ | 5.033 | <0.001 | $-2.940 \pm 1.255$ | 2.342 | 0.022 |
| Tri | BM | 0.257 | $0.015 \pm 0.003$ | 4.664 | <0.001 | | | |
| | BM + GC | 0.293 | $0.015 \pm 0.003$ | 4.711 | <0.001 | $-2.044 \pm 1.149$ | 1.778 | 0.080 |
| Tetra | BM | 0.290 | $0.019 \pm 0.004$ | 5.072 | <0.001 | | | |
| | BM + GC | 0.393 | $0.019 \pm 0.003$ | 5.382 | <0.001 | $-4.074 \pm 1.257$ | 3.241 | 0.002 |
| Penta | BM | 0.123 | $0.011 \pm 0.004$ | 2.972 | 0.004 | | | |
| | BM + GC | 0.205 | $0.011 \pm 0.004$ | 3.050 | 0.003 | $-3.272 \pm 1.296$ | 2.525 | 0.014 |
| Hexa | BM | 0.254 | $0.013 \pm 0.003$ | 4.634 | <0.001 | | | |
| | BM + GC | 0.268 | $0.013 \pm 0.003$ | 4.620 | <0.001 | $-1.129 \pm 1.049$ | 1.076 | 0.286 |

**Notes.**

Key to symbols: All, all imperfect microsatellites; Di, Tri, Tetra, Penta, Hexa, means imperfect microsatellites with different repeat type; BM, Body mass; GC, GC content.

general description of the basic characteristics of microsatellites, we particularly focused on comparing the motif mismatches of imperfect microsatellites to body mass across bird species in a phylogenetic context.

We found a negative relationship between body mass and the ratio of frequency of imperfect repeats (mismatch = 1) to perfect (mismatch = 0) ones with the same length 20bp among the species. Moreover, it is known that mutations in microsatellites shorter than a critical length are generally gain or loss of single repeat units which cannot disturb the repeat tract (*Buschiazzo & Gemmell, 2006*). Whereas when it reached a critical length, mismatch was introduced, a perfect microsatellite became imperfect. Here, our result implied that the introduction of motif mismatches in imperfect microsatellites is significantly associated with the nature of point mutation in microsatellites. In small-bodied species, since more perfect microsatellites suffer from the introduction of mismatches due to the higher mutation rate, a larger number of imperfect microsatellites relative perfect ones can be observed.

We observed that the microsatellites harbouring mismatches higher than expected have lower abundance than that carrying mismatches lower than expected. Consistent with this result, we also found that the microsatellites ≥30 bp and <3 mismatches have lower abundance than that ≥30 bp and >3 mismatches, indicating that mismatches of motifs is a key determinant leading to a paucity of long imperfect microsatellites in the genome. That is to say, mismatches would stabilize the repeat array and impede the further expansion. When the extent of mismatches reached saturation point, the repetition pattern is interrupted, leading the microsatellites to degeneration and death. (*Taylor, Durkin & Breden, 1999*; *Harr & Schlotterer, 2000*; *Yamada et al., 2002*; *Vowles & Amos, 2006*). Although the exact

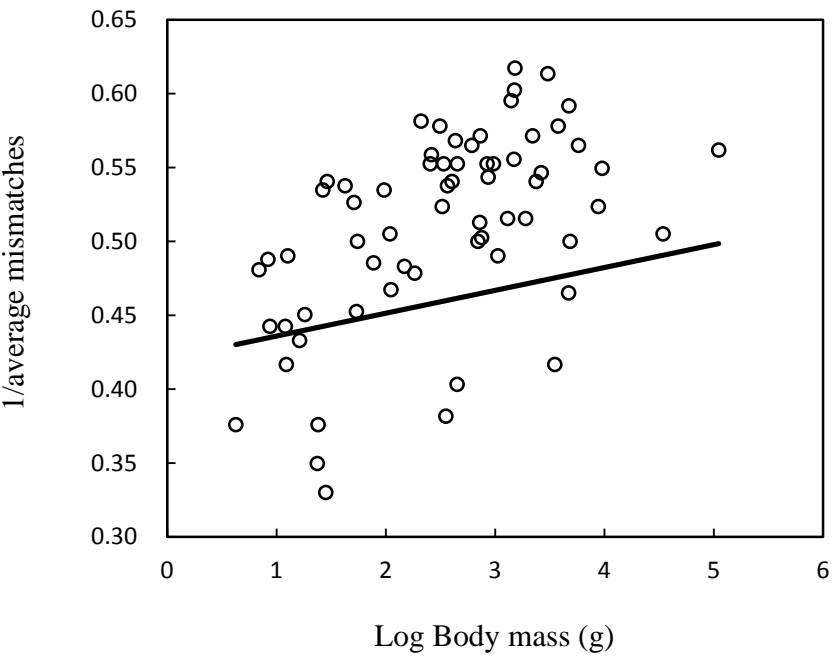

**Figure 1** Regression scatterplot of the inverse of the average mismatches of imperfect SSRs on the log of body mass in whole genome scale.

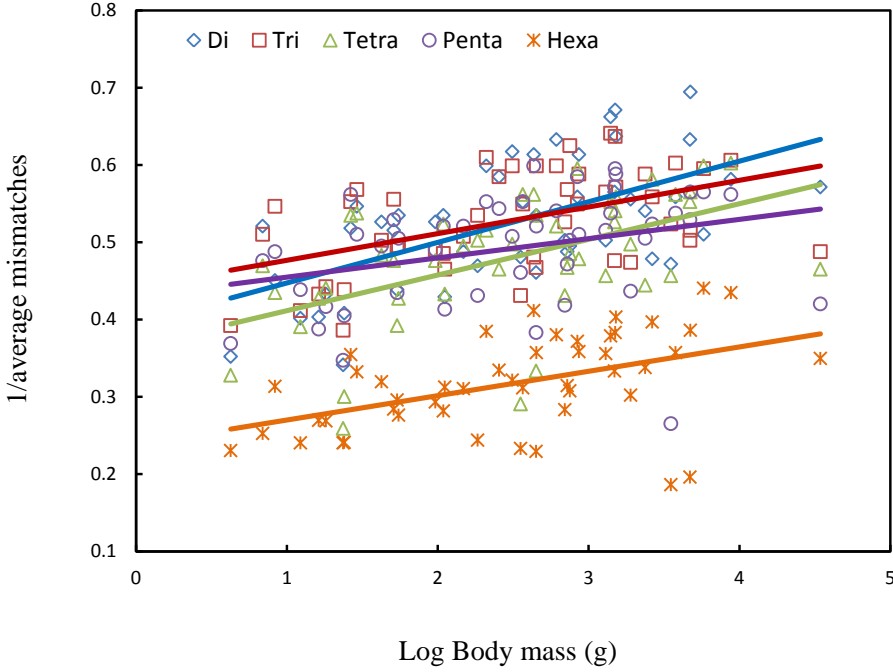

**Figure 2** Regression scatterplot of the inverse of the average mismatches of imperfect SSRs in five classes of repeat type on the log of body mass.

details of death is still poorly understood, the relative number of older mismatches in an 'average' microsatellite is likely to reflect the mutability during its lifetime. It can be further confirmed by the finding that the average mismatches of imperfect SSRs decreases with increasing body mass.

Our results that higher average mismatches of imperfect SSRs in small-bodied species support a correlation between mutation rate and life history traits. The pattern is usually explained by a generation length model, where smaller species evolve faster due to higher number of mutation opportunities per unit time (*Li et al., 1996*). In addition, body mass might affect the mutation rate through a link with metabolic rate and/or body temperature, which can directly change the mutation probability in a round of DNA replication (*Mindell et al., 1996*). Apart from these two key hypotheses, a rising hypothesis which proposes mutation rates are influenced by heterozygosity (*Amos, 2010*) can better explain the intrinsic correlation of motif mismatches with body mass. Smaller species have larger number of imperfect microsatellites which has been demonstrated by our data ($\beta \pm SE = -0.008 \pm 0.002$, $t = 4.008$, $P < 0.001$; $R^2 = 0.203$). Meanwhile, more heterozygous sites at these imperfect microsatellites can be expected. Recognition and 'repair' of heterozygous sites during synapsis will cause additional rounds of DNA replication which in turn provide more opportunities for mutations (*Amos, 2011*) and introduce more motif mismatches at imperfect microsatellite sites. Therefore, a negative relationship between body mass and motif mismatches can be observed. We suggest that heterozygote instability hypothesis, which is supported by increasing evidence (*Drake, 2007*; *Masters et al., 2011*; *Amos, 2013*; *Amos, 2016*), could provide a potential link between body mass and motif mismatches. However, further studies are needed in order to examine carefully whether homologous imperfect microsatellites are generally more prone to introduce mismatches in smaller species with a detail comparison between sister species.

## CONCLUSIONS

In conclusion, the present study is the first effort to explore the motif mismatch patterns of microsatellites in birds on a genomic level. The results we obtained from this study provide support for the long-standing correlation between mutation rate and life history traits and suggest that a negative body mass trend in mutation rate may be a general pattern of avian molecular evolution.

## ACKNOWLEDGEMENTS

We thank Xin Lu, Hongtao Xiao, Guoyue Zhang, Changcao Wang, Qingchen Zhang and Juanjuan Rao for data collection, statistical advice and insightful discussions. We also thank William Amos and Andrew Clarke for helpful suggestions and two anonymous referees for comments on earlier versions of this manuscript.

### Funding

The authors received no funding for this work.

### Competing Interests

The authors declare there are no competing interests.

### Author Contributions

- Haiying Fan conceived and designed the experiments, performed the experiments, analyzed the data, prepared figures and/or tables, authored or reviewed drafts of the paper.
- Weibin Guo performed the experiments, contributed reagents/materials/analysis tools, authored or reviewed drafts of the paper.

### Data Availability

The raw data is provided in the Supplemental Files.

### Supplemental Information

Supplemental information for this article can be found online at http://dx.doi.org/10.7717/peerj.4495#supplemental-information.

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
