# Peer review of "A genome-wide investigation of microsatellite mismatches and the association with body mass among bird species"

_PeerJ, doi:10.7717/peerj.4495_

## Round 0.1 · original submission · Major Revisions

This manuscript has been seen by two reviewers who suggested ‘major’ and ‘minor’ revisions. I think this is about right. If there was a box for ‘medium’ revision I would tick it. Since there is no such box, I will have to go for major, in that there is a reasonable amount of extra work required to address the Referees’ requests.

I agree with both Referees that the paper is generally nice but could be a lot nicer with the addition of extra information / data as requested. I realise it is tedious to go back and add more data, but it is also true that quite a few more high quality genomes have been published since the 2014 cut-off. My impression is that the recent genomes are substantially better quality. Consequently, while I would not insist on adding all the most recent genomes, I would like to see some added, as much as anything to ask whether the large jump is sequence quality affects the conclusions. Of course, with an efficient pipeline it should be quite easy to add all those that are both freely available and where accessible data exist for the biological parameters. I am rather less convinced of the need to fit clutch size and longevity, and I see these as being nice but very much optional additions.

Please pay close attention to the Referees’ other suggestions which are generally very constructive. I look forward to seeing the revised manuscript.

Reviewer 1 ·

Basic reporting

This is a well written article and represents a nice extension to our knowledge of microstatellites. I have listed a variety of suggestions and issues below. Most of these should be easy for the authors to fix.

lines 15-18: This sentence reads awkwardly perhaps change "except for" on line 17 to "besides".

line 22: should probably cite this paper:
Adams R.; H. Blackmon; J. Reyes-Velasco; D. Schield; D. Card; A. Andrew;
N. Waynewood; T. Castoe. Microsatellite landscape evolutionary dynamics across
450 million years of vertebrate genome evolution. Genome 59:5, 295-310.

They documented abundance for all types of 2-6mer SSRs in 5 species of birds. Actually might be nice to say whether your finding on totals match up with these numbers from their supplement:

Species Loci per Mbp 2mer 3mer 4mer 5mer 6mer All 2-6mer
Gavialis gangeticus 65.41 46.32 89.44 28.12 9.26 238.54
Taeniopygia guttata 33.06 49.15 86.86 31.30 9.78 210.14
Ficedula albicollis 33.96 49.36 113.22 37.60 16.97 251.11
Gallus gallus 42.35 59.52 121.68 39.09 9.09 271.74
Meleagris gallopavo 36.92 51.82 112.43 31.76 8.38 241.31

line 24-32 This may be a bit confusing some readers may think that SMM is an alternative to replication slippage. Maybe avoid this by changing your text on line 29-30 to: "Several mathematical models of microsatellite evolution have been proposed to represent the mutation processes...."

line 34 should cite the Adams paper above here as well

line 58 provide a citation for this software

line 82 you give a link to the website for the avian phylogenomics project as the source of your tree that many of your analyses depend on. I was unable to locate or find this tree to evaluate its appropriateness for your study. Please provide the tree as supplemental material or provide a doi to the data that you are reusing for instance at Dryad. This is concerning especially with regard to PGLS analyses since the tree structure can have large impacts on the significance of your results. Furthermore, since you did your work in R please provide a script that performs the analyses you describe.

line 114 I am not sure what this sentence is communicating.

line 138 This should be tested statistically

line 146-147: This is one example of a phrasing that I found difficult to keep straight:

"The correlation analysis revealed that microsatellites harboring more mismatches than expected are significantly related with that carrying fewer mismatches than expected"

At first read, this seems as though your saying something not possible. Upon further reading and reflection, I think that maybe a better way to say this would be:

"The correlation analysis revealed that microsatellites harboring more mismatches than expected are present in far fewer copies than expected"

If this isn't what you mean then I don't understand it and I think readers may have similar troubles. This applies later when you discuss this result in lines 183-184.

More broadly throughout the paper I found it difficult at times to keep track of the meaning of the measures that you are using. I would recommend putting a parenthetical reminder to the reader in a few places for instance in the discussion when you start talking about some of the measures like average mismatch.

line 166-167 I am not sure what is you think is propitious (using the defaults, using SciRoKo,..) regardless please explain more explicitly.

line 215-216 This should be tested rather than stated based on a visual examination. Some of the slopes look very similar and seem as though they may not be significantly different.

figure 1 you don't have a variable body mass effect. Proper description would be:

Regression scatterplot of the inverse of the average mismatches of
imperfect SSRs on the log of body mass.

figure 2 you aren't regressing onto mismatches nor body mass effect... Usually, the description of a regression is the opposite of the way you are presenting it. What you have plotted would be described as:

Regression scatterplot of the inverse of the average mismatches of
imperfect SSRs in five classes on the log of body mass.

Experimental design

Problems with the ability to replicate are described above with lack of tree source.

Validity of the findings

Seems sound without the phylogeny some items cannot be evaluated.

Reviewer 2 ·

Basic reporting

The manuscript submitted by Fan and Guo aims to test the link between mismatches in microsatellites and body and 48 of the birds species for which the genome is available.

The manuscript is well written and easy to understand for a non native English speaker (which I am). Yet, I think that some part of the manuscript are a bit difficult to understand (lines 154, 171, 183) and could be improved by a native English speaker. The world 'Moreover' is probably repeated too often (lines 136-144).

The overall structure of the manuscript is standard for scientific articles.

To help the understanding of the manuscript, I think that some of the table added in Supplementary Material (e.g. Table S2) could be added in the main text.

Experimental design

The theoretical background is well presented and the underlying question well presented. The results are interesting and suggest that small birds have a higher number of mismatches in microsatellites than larger birds.

Why did the authors not use sequences from genomes that have been published since 2014 as their approach (SciRoKo) could have allowed to do it 'fairly easily' ?

This is especially pertaining as some these newly published genomes have higher coverage than some of the 48 genomes published in 2014.

Body mass is a fairly simple trait to find but I wonder why it was the only one life history trait that was analysed. Other traits could be have been easily tested using the same hypothesis and/or controlled for (clutch size, longevity)

Validity of the findings

The statistical analyses seem to have been well performed, although i think that other easy to gather life history traits could have been added (clutch size, longevity) and controlled for in the analyses. Given the limited number of species analyses, this could be easily gathered in a short period of time and will strengthen the results.

Additional comments

No comment

---

## Round 0.2 · accepted · Accept

This manuscript has been appreciably improved and now seems ready for publication.

Reviewer 1 ·

Basic reporting

The authors have made significant changes to the manuscript and addressed my concerns. I feel that the revised manuscript is acceptable.

Experimental design

The authors have expanded to include more data and dealt with the minor statistical issues I had suggested.

Validity of the findings

Supplemental data is now available (i.e. the phylogeny) that will make their research reproducible by other labs and I see no issues with the applied approaches.

Additional comments

I appreciate your efforts to address the issues that I noted in the paper. I feel that the revised manuscript will be accessible to a wider audience.